# Effects of Mindfulness-Based Interventions in Children and Adolescents with ADHD: A Systematic Review and Meta-Analysis of Randomized Controlled Trials

**DOI:** 10.3390/ijerph192215198

**Published:** 2022-11-17

**Authors:** Yi-Chen Lee, Chyi-Rong Chen, Keh-Chung Lin

**Affiliations:** 1School of Occupational Therapy, College of Medicine, National Taiwan University, Taipei 100, Taiwan; 2Department of Psychiatry, Kaohsiung Chang Gung Memorial Hospital, Chang Gung University College of Medicine, Kaohsiung 833, Taiwan; 3Division of Occupational Therapy, Department of Physical Medicine and Rehabilitation, National Taiwan University Hospital, Taipei 100, Taiwan

**Keywords:** ADHD, systematic review, meta-analysis, mindfulness-based intervention

## Abstract

(1) Objectives: Mindfulness-based interventions have been receiving more attention in research for children with attention deficit hyperactivity disorder (ADHD). This systematic review and meta-analysis was conducted to synthesize the findings of randomized controlled trials of mindfulness-based interventions for children with ADHD. (2) Methods: A systematic review and meta-analysis of studies published in PsycINFO, PubMed, and Google Scholar was completed from the earliest available date until August 2022. (3) Results: The systematic review included 12 studies that met the inclusion criteria, and the meta-analysis included 11 studies. The overall effect sizes were *g* = 0.77 for ADHD symptoms, *g* = 0.03 for externalizing behavior problem, *g* = 0.13 for internalizing behavior problem, *g* = 0.43 for mindfulness, and *g* = 0.40 for parental stress for children with ADHD. (4) Conclusion: The results of this systematic review highlight the possible benefits of mindfulness-based interventions for children with ADHD.

## 1. Introduction

Attention deficit hyperactivity disorder (ADHD) is one of the most prevalent neurodevelopmental disorders, affecting more than 5% children worldwide [1,2]. Inattention and/or hyperactivity–impulsivity are two primary symptoms in children with ADHD [3]. Compared with peers without ADHD, children with ADHD have higher risk for multiple adverse outcomes, including poorer social and academic functioning [4] and increased mental health problems [5]. ADHD is also associated with substantial societal and family burden, particularly impacting parents [6]. Problem behaviors in children with ADHD may cause hostility and defiant behavior toward parents, leading to higher parenting stress [6]. Compared with parents of children without ADHD, parents of children with ADHD had significantly higher parenting stress and more dysfunctional parent–child interaction patterns [7,8,9]. Negative parenting behaviors were associated with poorer child academic, social, and emotional functioning and the formation of a coercive negative cycle between parents and children with ADHD [10]. 

Stimulant medication is the first-line treatment for ADHD and is effective for reducing ADHD symptoms. However, children with ADHD who are treated with medication still have poorer prognosis than those without ADHD [11]. In addition, adverse effects, such as insomnia and loss of appetite, are common in children with ADHD when they take stimulant medications [12].

Behavioral intervention has better outcomes as rated by those who are aware of treatment status, but is less effective for those rated by someone blinded to the intervention status [13]. Traditional behavioral therapy is implemented by adults (e.g., parents or teachers) providing rewards or aversive consequences for children’s behavior, and the maintenance effect is diminished [13]. Therefore, evidence-based and non-pharmaceutical interventions to improve outcomes for children with ADHD are needed.

Mindfulness-based interventions are a promising approach and have received increasing attention in the field of mental health [14]. Mindfulness involves focusing on the present moment without judgement or reaction [15]. Mindfulness meditations include choosing a point of focus, such as the breath, and focusing attention to that point with sustained attention [16]. In addition to the mindfulness meditation aspects, mindfulness movement practice, such as Yoga, emphasizes interoceptive, proprioceptive, and kinesthetic aspects of the movement experience [17]. The core and shared goals of a mindfulness-based intervention include strengthened awareness and a more integrated sense of self-accomplishment through the attention regulation process [18].

Mindfulness-based interventions are one of the best options to address the deficits associated with ADHD. Mindfulness focuses on the present moment with attention and emotion regulation, the very regulatory capacities that are impaired in ADHD [19]. ADHD and mindfulness go through similar processes. Children with ADHD have difficulty in sustained attention and impulse control, whereas mindfulness builds the regulatory capacity to observe external and internal stimuli without reacting to them [20].

Despite the potential benefits of mindfulness-based interventions in the management of ADHD, previous systematic reviews cannot offer definitive conclusions about the effectiveness of mindfulness-based interventions for children with ADHD because the methodological quality of the studies reviewed is low [21]. To overcome this gap, this meta-analysis systematically summarized studies of randomized controlled trials focusing on mindfulness-based interventions for children with ADHD.

## 2. Method

### 2.1. Search Strategy

The PubMed, PsycINFO, CINAHL, Web of Science, Google Scholar, and Scopus databases were searched for studies published from 1970 to 2022. The search strategy comprised the following Medical Subject Heading terms: yoga or mindfulness or mindful awareness or meditation and child or children or adolescent or adolescence and ADHD or inattention or hyperactivity/impulsivity. Studies were organized in the references list to assist the reviewers to perform study selection. We first identified studies by title and abstract to assess whether they met the inclusion criteria, and we also searched the reference lists of relevant systematic reviews for additional articles.

### 2.2. Inclusion Criteria

Studies were included in this systematic review if they randomized participants to the intervention or control group; if they included interventions with a focus on yoga, meditation, and/or mindfulness-based techniques such as mindful eating, mindful walking, or Taichi; if study participants had a diagnosis of ADHD; and if study participants were children/adolescents aged between 5 and 18 years. Yoga and/or mindfulness interventions that were combined with other approaches, such as parent training and non-specified relaxation techniques, were included.

### 2.3. Methodological Quality Assessment

We used the Physiotherapy Evidence Database (PEDro) Scale to assess the methodological quality of the included studies. The PEDro scale has 11 items. The PEDro scale criteria are: (1) if eligibility criteria are specified; (2) if participants are randomly allocated; (3) if allocation is concealed; (4) if the baseline comparison is similar between groups; (5) if participants are blinded; (6) if therapists are blinded; (7) if assessors are blinded; (8) if there is greater than 85% follow-up for at least one key outcome; (9) if intention-to-treat analysis is applied; (10) if there is between-group statistical comparison for at least one key outcome; and (11) if there are point estimates and measures of variability for at least one key outcome. A study will receive a point when the criterion of the item is met. Item 1 is not given a point. The total score of the PEDro is the sum of the points of items 2 to 11, for a maximum score of 10. A PEDro total score of 9 to 10 is considered as excellent, 6 to 8 as good, 4 to 5 as fair, and a score of less than 4 is considered as poor [22]. The first and second authors assessed studies independently. Discrepancies were resolved through discussion.

### 2.4. Data Extraction

Data extracted from each study included study design, participant characteristics, intervention characteristics, and outcome measures. The first author extracted and tabulated the relevant data from the studies, and the second author double-checked the information.

### 2.5. Statistical Analysis

Comprehensive Meta-analysis version 3 was used to calculate effect size estimates of the differences in performance scores (mean and standard deviations) [23]. Relevant quantitative data from each study were analyzed to calculate the intervention effect size. The reported means, standard deviation, and sample size from control and experimental groups at the post-intervention time point were used to calculate effect size *g*. Based on the guidelines suggested by Cohen [24], effect sizes were classified as small (0.2), medium (0.5), and large (0.8). Effect size estimates were separately calculated for ADHD symptoms, externalizing behavioral problem, internalizing behavioral problem, child’s mindfulness, and parenting stress. A random-effect model was used.

### 2.6. Test of Heterogeneity and Analysis of Moderator

The effect sizes of the studies might vary by chance [25]. A heterogeneity analysis was used to examine whether sampling error could explain the variance in a set of effect sizes. If the heterogeneity analysis revealed that the variance in the effect sizes was greater than expected by chance, other sources of variance, such as study characteristics or design variables, were examined. *Q* statistics were used to test the presence or absence of heterogeneity. *I*^2^ was calculated to determine the proportion of observed variance that reflects a real difference in effect size [25]. *I*^2^ values of 25%, 50%, and 75% are interpreted as low, moderate, and high proportions of real differences in effect size, respectively [26].

To estimate the effect of categorical (e.g., no intervention control versus active control) and continuous variables (e.g., methodological rating) on the effect of mindfulness-based intervention, we performed sub-group analysis and meta-regression analyses, respectively. The possibility of publication bias was examined. Funnel plot, the Begg and Mazumdar test, and Duval and Tweedle’s trim and fill method were adopted to impute the value of missing studies.

## 3. Results

### 3.1. Descriptions of Studies

We identified 936 articles from the databases (See Figure 1). Two additional studies found from the references of identified systematic reviews were included. We excluded 363 duplicated articles and screened 573 articles by title and abstract. Of these, 506 articles were excluded. A total of 67 papers were retrieved to screen for further details, of which 12 met the inclusion criteria. However, one study [27] that met the inclusion criteria did not provide data for the meta-analysis. We contacted the authors but did not receive a response. Therefore, there were 12 studies for systematic review and 11 for meta-analysis.

Study authors spanned a number of countries or regions, including Iran [28,29], Spain [30,31], Italy [32,33,34], the United States [27,35], Australia [36], Romania [37], and Hong Kong [38]. All studies except one [36] were published in or after 2017.

### 3.2. Participants

As detailed in Table 1, the age varied across the studies. Two studies recruited adolescents [28,29], whereas the others recruited children aged between 5 and 12 years [27,30,31,32,33,34,35,36,37,38,39]. Most studies recruited significantly higher percentages of boys with ADHD than girls [27,30,31,32,33,34,35,36,37,38,39], whereas two studies recruited all female adolescents with ADHD as participants [28,29].

### 3.3. Intervention

Three types of mindfulness-based intervention were identified: yoga intervention [27,36], mindfulness-based psychological intervention [28,29,30,31,32,35,36,37,38,39], and meditation training [33,34]. Five studies provided mindfulness-based training to both children or adolescents with ADHD and to their parents [30,31,32,38,39]. In addition to group programs for children with ADHD, four studies provided additional mindfulness-based groups for parents of children with ADHD [31,32,38,39]. One study interviewed children with ADHD and their parents before the start of the program and before the end of each session to increase the program compliance of children with ADHD and their parents [30].

In addition to experiential intervention in groups, most programs emphasized the importance of the implementation of home programs [27,28,29,30,31,32,33,34,36]. Homework exercises were provided, checked, and discussed for implementation at home in a structured group intervention [28,29,30,31,32,33,34,36,39].

Lengths of individual sessions ranged from 20 to 90 min, with most sessions lasting 90 min [28,32]. The length of the entire intervention ranged from only one session to 8 weeks, with most interventions spanning 8 weeks [29,31,33,34,39]. The most common intervention dose was 12 h [28,29,31,32,39]. The dose of the regimen [36] was as high as 20 h.

### 3.4. Control Group

All of the reviewed studies randomized participants to an intervention or control group, although some studies provided no information regarding the randomization procedures [28,29,30,31,35,36]. Five studies used a waiting list control group [28,29,31,32,38]. Two studies used treatment as usual [39] or standard care [30] as control groups. Four studies used an active control group, such as a listening task [37], behavioral therapy [35], cooperative activities [36], or an emotional education program [33,34].

### 3.5. Outcome Measures

This review targeted five primary outcomes, namely, children’s ADHD symptoms, children’s internalizing problems, externalizing problems, mindfulness, and parental stress. Seven studies measured the symptoms of ADHD [28,31,32,33,34,36,38,39] with SNAP-IV, an abbreviated version of the Swanson, Nolan, and Pelham (SNAP) Questionnaire [28], the Conners’ Parent Rating Scale–Revised [33,34,36,39], the Conners’ Teacher Rating Scale–Revised [36], the Strengths and Weaknesses of ADHD Symptoms and Normal Behaviors Rating Scale (SWAN) [38,39], the Strengths and Difficulties Questionnaire [32], Conners-3rd Edition [31], and the Child Behavior Checklist [34].

Children’s internalizing problems and externalizing problems were measured in nine studies with the Child Behavior Checklist [30,38], the Conners’ Parent Rating Scale-Revised [33,36], the Conners’ Teacher Rating Scale-Revised [36], the Difficulties in Emotion Regulation Scale [29], the Present Functioning Visual Analogue Scale [37], the Modified Overt Aggression Scale [32], the Strengths and Difficulties Questionnaire [32], and Conners-3rd Edition [31].

Children’s mindfulness was measured in four studies with the Mindful Attention Awareness Scale [28,35] and the Child and Adolescent Mindfulness Measure [32,39]. Parental stress was measured in two studies with the Parenting Stress Index-Short Form [31,38].

### 3.6. Quality Assessment

Twelve studies scored between 4 and 7 points from a possible total of 10 points on the PEDro Scale (Table 2), with an average score of 5.58. This suggests that the included studies were of moderate methodological quality. Patient blinding and therapist blinding were impractical to obtain in this systematic review.

### 3.7. Effects on ADHD Symptoms

Results from seven studies showed that the combined effect size of the mindfulness-based interventions on ADHD symptoms was *g* = 0.77 (95% CI = 0.22, 1.33, *Z* = 2.72, *p* = 0.006), suggesting that mindfulness-based interventions had a significant effect on the reduction of ADHD symptoms (Figure 2). The assessment of overall heterogeneity across the studies indicated the non-ignorable presence of heterogeneity (*Q* = 32.87, *df* = 6, *p* < 0.001). The *I*^2^ with 81.75% suggested high heterogeneity as a result of the high variability across the studies. The Begg and Mazumdar test revealed possible publication bias (Tau = 0.62, *p* = 0.03). There were two articles trimmed to the right mean using a random model according to the Duval and Tweedie’s trim and fill for ADHD symptoms. The Begg funnel plot showed the basically symmetric distribution (Figure 3).

Two studies measured the follow-up effect of ADHD symptoms. The overall effect size was calculated at *g* = 0.34 (95% CI = −0.42, 1.09; *Z* = 0.87, *p* = 0.38) (Table 3). The heterogeneity test of effect size was not significant (*Q* = 3.37, *df* = 1, *p* = 0.07, *I*^2^ = 70.29%).

### 3.8. Effects on Externalizing Behavioral Problem

A trivial effect on externalizing behavioral problem was observed with *g* = 0.03 (95% CI = −0.17, 0.23; *Z* = −0.27, *p* = 0.76), as presented in Figure 4. The heterogeneity test results showed nonsignificant heterogeneity within results with *Q* = 1.28 (*df* = 6, *p* = 0.97) and *I*^2^ = 0%. The Begg and Mazumdar test revealed no publication bias (Tau = 0.24, *p* = 0.23). There were no missing articles to the right mean using a random model according to the Duval and Tweedie’s trim and fill for externalizing behavioral problem. The Begg funnel plot showed the basically symmetric distribution (Figure 5).

Two studies measured the follow-up effect of externalizing behavioral problem. The overall effect size was calculated at *g* = 0.18 (95% CI = −0.30, 0.66; *Z* = 0.74, *p* = 0.46). The heterogeneity test of effect size was not significant (*Q* = 1.53, *df* = 1, *p* = 0.22, *I*^2^ = 34.73%).

### 3.9. Effects on Internalizing Behavioral Problem

Based on seven studies, the combined effect size of mindfulness-based intervention on internalizing behavioral problem was *g* = 0.13 (95% CI = −0.24, 0.50; *Z* = 0.71, *p* = 0.48), as presented in Figure 6. The heterogeneity test results showed significant heterogeneity within results with *Q* = 19.12 (*df* = 6, *p* = 0.004) and *I*^2^ = 68.63%. The Begg and Mazumdar test revealed no publication bias (Tau = 0.14, *p* = 0.33). There were no missing articles to the right mean using a random model according to the Duval and Tweedie’s trim and fill for internalizing behavioral problem. The Begg funnel plot showed the basically symmetric distribution (Figure 7).

The follow-up effect on internalizing behavioral problem was *g* = −0.27 (95% CI = −1.15, 0.62; *Z* = −0.59, *p* = 0.55). The heterogeneity test results showed significant heterogeneity within results with *Q* = 7.55 (*df* = 1, *p* = 0.006) and *I*^2^ = 86.76%.

### 3.10. Effects on Child’s Mindfulness

Based on four studies, the combined effect size for child’s mindfulness was *g* = 0.43 (95% CI = −0.27, 1.13; *Z* = 1.20, *p* = 0.23) (Figure 8). The heterogeneity test results showed significant heterogeneity within results with *Q* = 19.30 (*df* = 3, *p* < 0.001) and *I*^2^ = 84.45%. The follow-up effect on child’s mindfulness was *g* = −0.17 (95% CI = −0.58, 0.25; *Z* = −0.80, *p* = 0.42). The Begg and Mazumdar test revealed possible publication bias (Tau = 1.00, *p* = 0.02). There was one article trimmed to the right mean using a random model according to the Duval and Tweedie’s trim and fill for child’s mindfulness. The Begg funnel plot showed the basically symmetric distribution (Figure 9).

### 3.11. Effects on Parental Stress

Based on two studies, the combined effect size for parental stress was *g* = 0.40 (95% CI = −0.42, 1.23, *Z* = 0.96, *p* = 0.34) (see Figure 10). The heterogeneity test results showed significant heterogeneity within results with *Q* = 4.03 (*df* = 1, *p* = 0.045) and *I*^2^ = 75.18%.

### 3.12. Subgroup Analysis

In the subgroup analysis of no intervention or an active control group for ADHD symptoms, the effect size for a no intervention control group was in the large range (g = 0.83 [95% CI = 0.13, 1.53], *Z* = 2.36, *p* = 0.02), and the effect size for the active control group was moderate to large (*g* = 0.70 [95% CI = 0.61, 1.34], *Z* = 2.15, *p* = 0.03). In addition, high heterogeneity was observed in the no intervention control groups (*Q* = 30.41, *df* = 4, *p* < 0.001, *I*^2^ = 86.85%) for ADHD symptoms.

In the subgroup analysis of no intervention or active control groups for internalizing behavioral problem, the effect size for a no intervention control group was in the small range (*g* = 0.19 [95% CI = −0.27, 0.66], *Z* = 0.82, *p* = 0.42), and the effect size for an active control group was close to zero (*g* = 0.005 [95% CI = −0.62, 0.63], *Z* = 0.01, *p* = 0.99).

### 3.13. Meta-Regression Analyses

The meta-regression results showed that the effect size of ADHD symptoms was significantly moderated by the child’s age (β = 0.24, SE = 0.11, *p* = 0.03), suggesting a larger mindfulness-based intervention effect was associated positively with older children. In addition, the meta-regression results showed that overall effect size was significantly moderated by the child’s sex (β = −0.03, SE = 0.001, *p* = 0.008) but with a very small coefficient. We found that the methodological rating had a small and negative association with effect sizes of ADHD symptoms (β = −0.27, SE = 0.23, *p* = 0.23).

The meta-regression results showed that the effect size of internalizing behavioral problem was moderated by the child’s age (β = 0.05, SE = 0.10, *p* = 0.58). In addition, the meta-regression results showed that overall effect size was moderated by the child’s sex (β = −0.008, SE = 0.008, *p* = 0.31) but with a very small coefficient. We found that the methodological rating had a small and negative association with the effect sizes of internalizing behavioral problem (β = −0.03, SE = 0.25, *p* = 0.93).

### 3.14. Sensitivity Analysis

We performed a sensitivity analysis by removing included articles one at a time to determine the impact of every single included study on the obtained results. The corresponding Hedge’s *g* was altered during this procedure from 0.77 to 0.44 for ADHD symptoms, and from 0.43 to −0.02 for mindfulness, when the study of Abdolahzad and colleagues [28] was removed, indicating that the study of Abdolahzad and colleagues [28] was an outlier in the meta-analysis. Other corresponding Hedge’s *g* values did not alter and presented the stability of the findings in this meta-analysis.

## 4. Discussion

This is the first meta-analysis of randomized controlled trials to examine the effects of mindfulness-based interventions on improving ADHD symptoms, externalizing behavioral problem, internalizing behavioral problem, mindfulness, and parental stress in children with ADHD. Using a control group for comparison with a mindfulness-based intervention means that the influence of maturational changes in children was excluded because these will be common between the mindfulness-based intervention and comparative conditions. In addition, we used a randomized design to exclude the influence of systematic differences between groups.

This meta-analysis synthesized seven studies of mindfulness-based interventions and revealed a moderate-to-large effect size for ADHD symptoms in children with ADHD. The findings are similar with a previous meta-analysis [40] that examined the effect of mindfulness-based interventions on attention and hyperactivity in childhood. In general, children with ADHD randomly assigned to a mindfulness-based intervention showed improvements in inattentive and hyperactive–impulsive behavior relative to children with ADHD in the control groups. These findings indicate that a mindfulness-based intervention can have a positive effect on attention, hyperactivity, and impulse control in children with ADHD.

To the best of our knowledge, this is the first meta-analysis to examine the effect of mindfulness-based interventions on externalizing behavioral problem and internalizing behavioral problem in children with ADHD. Our results are consistent with a previous systematic review [21] that there was limited support for a reduction in internalizing and externalizing behavioral problem in children with ADHD after mindfulness-based intervention compared with the control group.

This meta-analysis found improvement that is small to moderate in magnitude for mindfulness after mindfulness-based interventions compared with control conditions. It is assumed that ADHD and mindfulness go through similar processes, with mindfulness focusing on the present moment with attention and emotion regulation, the prevalent deficits in children with ADHD [20]. In this meta-analysis, mindfulness-based interventions improved both ADHD symptoms and mindfulness. However, the findings of improvement in both ADHD symptoms and mindfulness do not imply a cause-and-effect relationship of these two outcomes or a mediating role for mindfulness in the improvement of ADHD symptoms in children with ADHD. Future studies may examine the mechanisms of mindfulness in the improvement of ADHD or inattentive symptoms after a mindfulness-based intervention in children with ADHD.

Larger effects on ADHD symptoms were associated with older children. It is possible that this was influenced by the symptoms of ADHD that change over developmental periods. When children with ADHD get older, inattentive symptoms become more obvious, and hyperactivity and impulsivity tend to decrease. Current research supports that mindfulness-based interventions increase attention capacity more than hyperactivity and impulsivity [41]. Adolescents or older children have more insight into ADHD symptoms, which could be another possible explanation for a larger effect in older children. Adolescents or older children have more capacity to integrate what they learn from a mindfulness-based intervention and have more reduction in symptoms after treatment than younger children with ADHD [42]. It is also possible that differences in mindfulness techniques may be more effective in different populations [42]. Mindfulness techniques that use visualization, active movement, or concrete techniques may be more effective for younger children. Future research may need to explore the effects of different mindfulness techniques for different populations.

There was very little influence of the mindfulness-based intervention at the follow ups, which were 1 to 6 months after the interventions. Although home practice was an important part of the mindfulness-based intervention programs, participants might not have continued practicing mindfulness exercises when the intervention was over. The effect might have diminished with time after the interventions ended. Future studies may need to explore the reasons and possible solutions for the reduced retention of intervention effects.

There were significant methodological limitations in the studies included in this meta-analysis. Several studies did not report the method of randomization or concealment of allocation before assignment. Most of the studies reported rates of attrition but did not report using statistical techniques to address incomplete data. Some studies did not blind the assessors who evaluated the outcomes. Although most studies allowed participants to remain on medication throughout the course of the studies, one study [27] did not report whether medication status was held constant throughout the intervention. Some studies [28,29,32,33,38] did not report the medication status of participants with ADHD. Of note, medication status may have confounded the study results. In addition, possible publication bias may exist with respect to ADHD symptoms and child’s mindfulness and raises concerns regarding the effect of mindfulness-based intervention for children and adolescents with ADHD.

## 5. Conclusions

This systematic review and meta-analysis included 12 randomized controlled trials that investigated mindfulness-based interventions for children with ADHD. The results show that mindfulness-based interventions are an effective intervention for reducing ADHD symptoms in children with ADHD. A large effect on ADHD symptoms is noted for older children or adolescents with ADHD. Further research is needed to study the effects of different types of mindfulness for different groups of age and sex.

## Figures and Tables

**Figure 1 ijerph-19-15198-f001:**
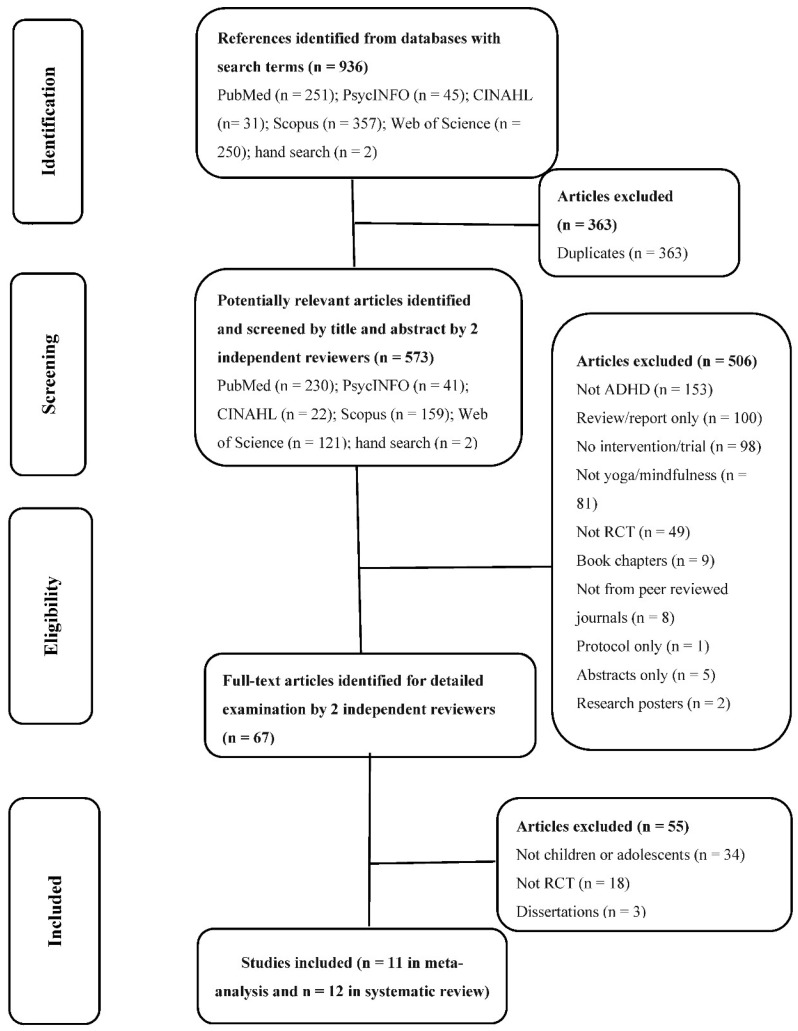
Flow diagram of the literature search.

**Figure 2 ijerph-19-15198-f002:**
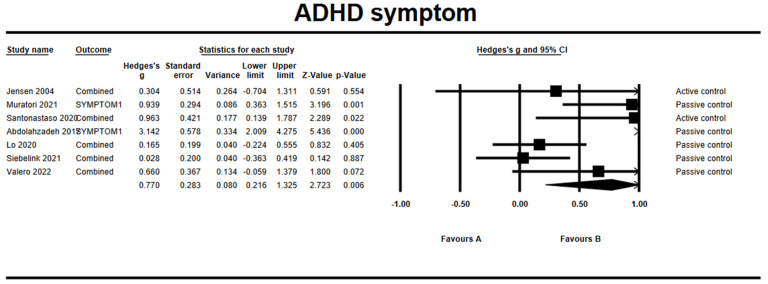
Forest plots of effect size for ADHD symptoms [28,31,32,33,36,38,39].

**Figure 3 ijerph-19-15198-f003:**
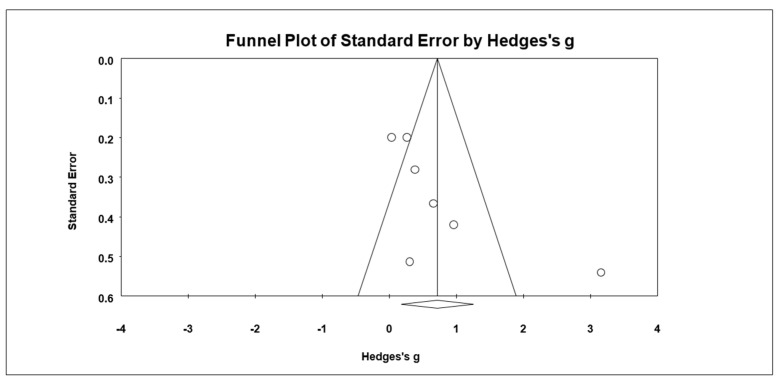
Funnel plot for ADHD symptoms.

**Figure 4 ijerph-19-15198-f004:**
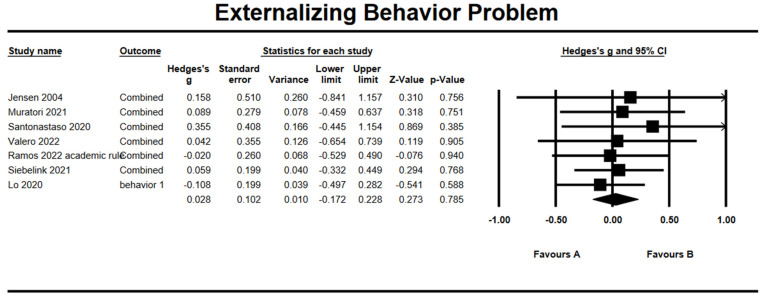
Forest plots of effect size for externalizing behavioral problem [31,32,33,35,36,38,39].

**Figure 5 ijerph-19-15198-f005:**
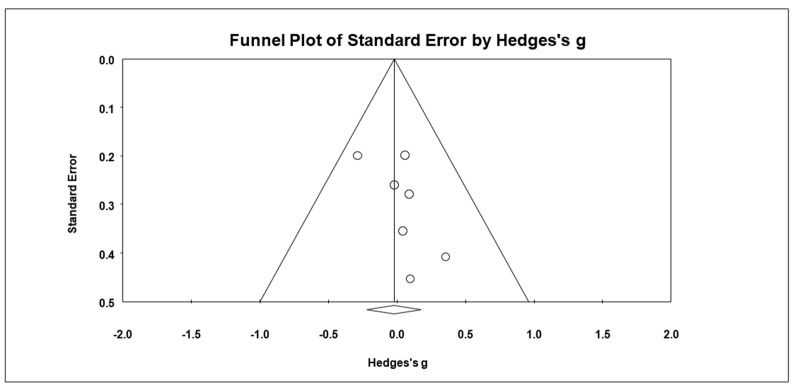
Funnel plot for externalizing behavioral problem.

**Figure 6 ijerph-19-15198-f006:**
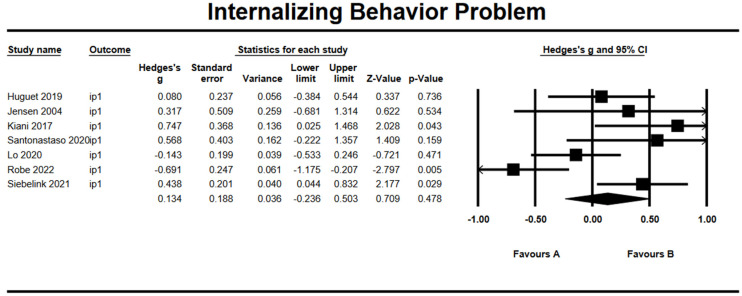
Forest plots of effect size for internalizing behavioral problem [29,30,33,36,37,38,39].

**Figure 7 ijerph-19-15198-f007:**
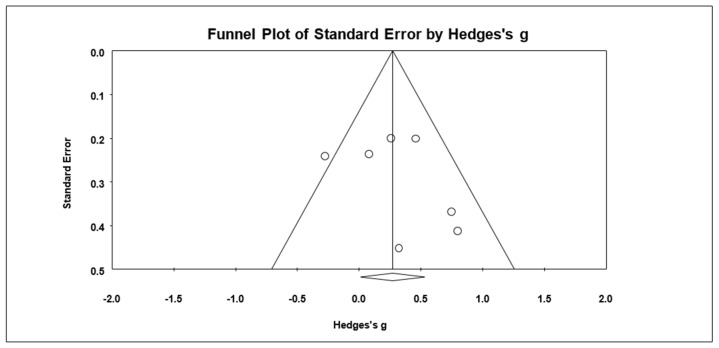
Funnel plot for internalizing behavioral problem.

**Figure 8 ijerph-19-15198-f008:**
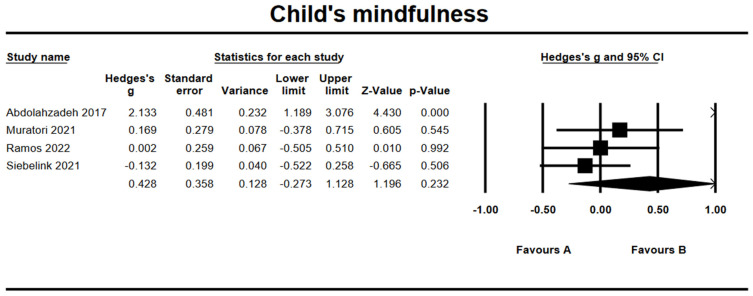
Forest plots of effect size for child’s mindfulness [28,32,35,39].

**Figure 9 ijerph-19-15198-f009:**
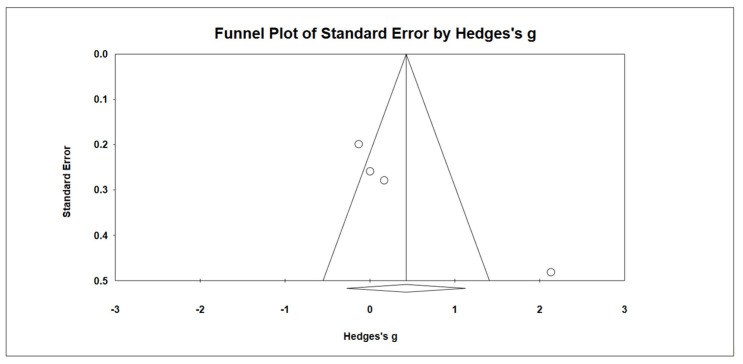
Funnel plot for child’s mindfulness.

**Figure 10 ijerph-19-15198-f010:**
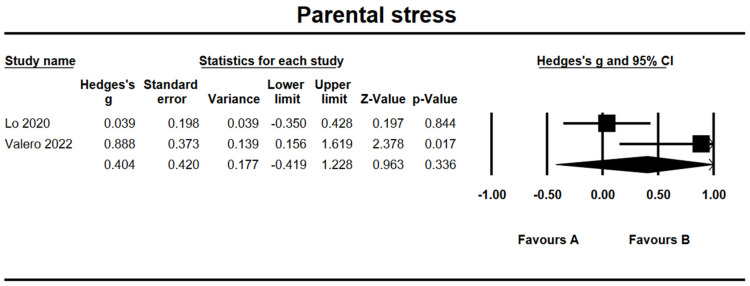
Forest plots of effect size for parental stress [31,38].

**Table 1 ijerph-19-15198-t001:** Characteristics of studies included in the meta-analysis.

				Experimental Group			Control Group						
Study (Year and Country of the Authors)	Age	ADHD Subtype (%)	Male (%)	N	Age, Mean (SD) or Range	Intervention	N	Age, Mean (SD) or Range, Years	Intervention	Follow- up	Outcome (ADHD Symptoms)	Informants	On ADHD Medication (%)
Abdolahzadeh et al. (2017, Iran) [28]	Adolescent	NR	0	13	Mean: 15.96	MBI, 90 min/session, 8 sessions	13	Mean: 15.96	Waitlist	No FU	SNAP-IV	Parent	NR
Cohen et al. (2018, USA) [27]	Children	NR	65.22	10	4.33 (0.58)	Yoga 30 min/session, twice/week, 6 weeks	11	3.83 (0.83)	Regular activities	6 weeks, 3 months	ADHD RS-IV preschool version	Parent Teacher	4.35
Huguet et al. (2019, Spain) [30]	Child Parent	C: 64.29% I: 31.43% Hy: 4.29%	72.86	34	8.79 (1.29)	MBI, 75 min/session, 8 sessions	36	8.81 (1.65)	Standard care	No FU	CBCL	Parent	0
Jessen et al. (2004, Australia) [36]	Children	C: 78.58% I: 22.42%	NR	11	10.63 (1.78)	Yoga, 60 min/session, 20 sessions	8	9.35 (1.70)	Cooperative activities	No FU	CPRS, CTRS	Parent Teacher	85.71
Kiani et al. (2017, Iran) [29]	Adolescent	NR	0	15	13.17 (.35)	Mindfulness mediation therapy (12 h/8 weeks)	15	13.42 (.73)	Waitlist	None			NR
Muratori et al. (2021, Italy) [32]	Children Parent	NR	100	25	8.75 (0.71)	MBI (90 min/session, 8 sessions)	25	9.05 (1.05)	Waitlist	No FU	CBCL, SDQ	Parent	NR
Ramos et al. (2022, USA) [35]	Children	C: 75.9% I: 22.4% Hy: 1.7%	93.2	29	9.2 (1.4)	MBI + BT 20 min/session, 3–4 sessions/week, 6-week summer camp	29	9.2 (1.4)	BT	No FU			63
Robe-Dobrean et al. (2022, Romania) [37]	Children	NR	62.9	35	11.66 (2.68)	MBI, 1 session	35	10.40 (2.77)	A listening task	4-week FU	ADHD-RS	Parent	10
Santonastaso et al. (2020, Italy) [33] Zaccari et al. (2022, Italy) [34]	Children	C: 85.71% I: 8.57% Hy: 5.71%	74.29	15	8.9 (1.3)	Mindful-oriented meditation, 3 times/week, 8 weeks	10	9 (1.2)	Emotional education program, 3 times/week, 8 weeks	2-month FU	CPRS-R:L CBCL	Parent	NR
Lo et al. (2020, HongKong) [38]	Child Parent	NR	83	50	6.24 (.87)	FBMI (Parent 9 h/6 weeks) (Child 8 h)	50	5.92 (.70)	Waitlist	None	SWAN	Observer	NR
Siebelink et al. (2021, Netherlands) [39]	Child Parent	NR	69.9	55	11.0 (1.8)	FMBI (12 h/8 weeks)	48	11.4 (1.8)	Treatment as usual	2-month FU	CPRS, SWAN	Observer	81
Valero et al. (2022, Spain) [31]	Child Parent	C: 56.7 I: 30 Hy: 13.33	76.7	15	10.33 (1.83)	FMBI (Parent 12 h/8 weeks) (Child 8 h/8 weeks)	15	11.6 (1.29)	Waitlist	6 mo	Conners—3rd Edition	Observer	56.67

ADHD: attention deficit hyperactivity disorder; NR: not report; MBI: mindfulness-based intervention; FU: follow-up; SNAP-IV: the Swanson, Nolan, and Pelham, version IV Scale; ADHD RS-IV: ADHD Rating Scale-IV; C: combined type; I: inattentive type; Hy: hyperactive type; CBCL: Child Behavior Checklist; CPRS: the Conners’ Parents Rating Scales, CTRS: the Conners’ Teachers Rating Scales; SDQ: Strengths and Difficulties Questionnaire; BT: behavioral therapy; CPRS-R:L: Conners’ Parent Rating Scales Long Version Revised; FBMI: family-based mindfulness intervention; SWAN: the Strengths and Weaknesses of ADHD Symptoms and Normal Behaviors Rating Scale.

**Table 2 ijerph-19-15198-t002:** Methodological quality assessment of included studies on the PEDro Scale.

Study	2	3	4	5	6	7	8	9	10	11	Total
Abdolahzadeh et al. (2017) [28]	1	0	1	0	0	0	1	0	1	1	5
Cohen et al. (2018) [27]	1	0	1	0	0	0	1	1	1	1	6
Huguet et al. (2019) [30]	1	0	1	0	0	0	1	0	1	1	5
Jensen et al. (2004) [36]	1	0	0	0	0	0	1	0	1	1	4
Kiani et al. (2017) [29]	1	0	1	0	0	0	1	0	1	1	5
Lo et al. (2020) [38]	1	0	1	0	0	1	1	1	1	1	7
Muratori et al. (2021) [32]	1	0	1	0	0	0	1	0	1	1	5
Ramos et al. (2022) [35]	1	0	1	0	0	0	1	0	1	1	5
Robe-Dobrean et al. (2022) [37]	1	0	1	0	0	1	1	0	1	1	6
Santonastaso et al. (2020) [33] Zaccari et al. (2022) [34]	1	1	1	0	0	1	1	0	1	1	7
Siebelink et al. (2021) [39]	1	0	1	0	0	1	1	1	1	1	7
Valero et al. (2022) [31]	1	0	1	0	0	0	1	0	1	1	5

Score 0 = absent/unclear, 1 = present.

**Table 3 ijerph-19-15198-t003:** Average effect sizes by types of outcome by times of measurement.

Time	ADHD Symptoms		Externalizing Problem		Internalizing Problem		Mindfulness		Parenting Stress	
	*k*	Hedge’s *g*	*k*	Hedge’s *g*	*k*	Hedge’s *g*	*k*	Hedge’s *g*	*k*	Hedge’s *g*
Post-treatment	7	0.77	7	0.03	7	0.13	4	0.43	2	0.40
Follow-up	2	0.34	2	0.18	2	−0.27	1	−0.17		

*k*: numbers of studies contributing to the average effect size *g.*

## Data Availability

Available upon request to the correspondence author.

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
