# Peer review of "Effects of Mindfulness-Based Interventions in Children and Adolescents with ADHD: A Systematic Review and Meta-Analysis of Randomized Controlled Trials"

_ijerph, 2022, doi:10.3390/ijerph192215198_

Round 1

Reviewer 1 Report

The manuscript highlights the importance of developing new strategies for ADHD treatment and refers to the advantages of mindfulness based interventions to treat ADHD symptoms. Authors perform a comprehensive search for randomized controlled trials addressing the effect of mindfulness-based interventions on ADHD symptoms in children and adolescents. Inclusion criteria for the studies, as well as description of the main characteristics of the studies are well done. Authors find a positive effect of mindfulness-based interventions on ADHD symptoms, being more effective in older children. Furthermore, in the studies where a follow up was performed no longer found the effect originally observed.

Authors perform a good analysis and state clearly the limitations of their study. It is, in my opinion, a good contribution to the field. There are only some points I believe should be added in order to have a more complete discussion

1.- The manuscripts mentions 2 tables, but I could not find them, neither on the manuscript nor in the supplementary material. Please add the tables

2.- At some point in the discussion authors mention that patients in some studies were under medication. Although it is difficult to analyze the influence of medication on the results observed, it would be useful for readers to know in which studies were the patients medicated and, if possible, whether they were under stimulant or non-stimulant medication. This could be included in a Table describing the main characteristics of each RCT.

3.- In the cases were a follow up was carried out, authors found very little influence of the intervention at the follow up. Please mention how long after the intervention was the follow up performed, and address in the discussion the possible causes of this finding.

4.- The current version of the manuscript mentions very little regarding the statistical analysis performed. Please include in the methods section the description of the statistical analyses, as well as the software used.

MINOR ISSUES

Line 226 says “Twelves studies” instead of “Twelve studies”

Author Response

The manuscript highlights the importance of developing new strategies for ADHD treatment and refers to the advantages of mindfulness based interventions to treat ADHD symptoms. Authors perform a comprehensive search for randomized controlled trials addressing the effect of mindfulness-based interventions on ADHD symptoms in children and adolescents. Inclusion criteria for the studies, as well as description of the main characteristics of the studies are well done. Authors find a positive effect of mindfulness-based interventions on ADHD symptoms, being more effective in older children. Furthermore, in the studies where a follow up was performed no longer found the effect originally observed.

Authors perform a good analysis and state clearly the limitations of their study. It is, in my opinion, a good contribution to the field. There are only some points I believe should be added in order to have a more complete discussion

1.- The manuscripts mention 2 tables, but I could not find them, neither on the manuscript nor in the supplementary material. Please add the tables

 Response: We have added three tables.

2.- At some point in the discussion authors mention that patients in some studies were under medication. Although it is difficult to analyze the influence of medication on the results observed, it would be useful for readers to know in which studies were the patients medicated and, if possible, whether they were under stimulant or non-stimulant medication. This could be included in a Table describing the main characteristics of each RCT.

 Response: Percentages of participants on medication were added in Table 1. We also addressed the issue of medication in the discussion:

“Although most studies allowed participants to remain on medication throughout the course of the studies, one study (Cohen et al., 2018) did not report whether medication status was held constant throughout the intervention. Some studies (Abdolahzadeh et al., 2017; Kiani et al., 2017; Lo et al., 2020; Muratori et al., 2021; Santonastaso et al., 2020) did not report the medication status of participants with ADHD. Of note, medication status may have confounded the study results.” (Page 12, Line 5-13)

3.- In the cases were a follow up was carried out, authors found very little influence of the intervention at the follow up. Please mention how long after the intervention was the follow up performed, and address in the discussion the possible causes of this finding.

  Response: The durations of follow-up were provided in Table 1. We also addressed the possible causes of the findings in the discussion.

“There was very little influence of the mindfulness-based intervention at the follow up, which were 1 to 6 months after the interventions. Although home practice was an important part of the mindfulness-based intervention programs, participants might not have continued practicing mindfulness exercises when the intervention was over. The effect might have diminished with time after the interventions ended. Future studies may need to explore the reasons and possible solutions for the reduced retention of intervention effects.” (Page 11, Line 32-38 )

4.- The current version of the manuscript mentions very little regarding the statistical analysis performed. Please include in the methods section the description of the statistical analyses, as well as the software used.

 Response: Statistical analysis and the software used were added in the methods section. (Page 4, Line 19 – Page 5, Line 8)

MINOR ISSUES

Line 226 says “Twelves studies” instead of “Twelve studies”

Response: It was changed to “Twelve studies”.

Reviewer 2 Report

This study investigates the effects of mindfulness-based interventions on ADHD symptoms. Although this is an important topic, the present study suffering lacks novelty, as several studies on similar topics have been published.

1.      Cairncross, M., & Miller, C. J. (2020). The effectiveness of mindfulness-based therapies for ADHD: A meta-analytic review. Journal of attention disorders, 24(5), 627-643.

2.      Chimiklis, A. L., Dahl, V., Spears, A. P., Goss, K., Fogarty, K., & Chacko, A. (2018). Yoga, mindfulness, and meditation interventions for youth with ADHD: Systematic review and meta-analysis. Journal of Child and Family Studies, 27(10), 3155-3168.

3.      Vekety, B., Logemann, H. A., & Takacs, Z. K. (2021). The effect of mindfulness-based interventions on inattentive and hyperactive–impulsive behavior in childhood: A meta-analysis. International Journal of Behavioral Development, 45(2), 133-145.

4.      Xue, J., Zhang, Y., & Huang, Y. (2019). A meta-analytic investigation of the impact of mindfulness-based interventions on ADHD symptoms. Medicine, 98(23).

So I do not recommend to publish it in the International Journal of Environmental Research and Public Health.

My comments are the following:

The authors mention to include not just mindfulness but yoga and Taiichi interventions. It would be better to treat them in separate analyses.

A big benefit is to provide methodological quality assessment was done. I recommend to compare the effect fas found in good vs fair quality studies.

Data extraction would have been better if it was performed by two independent coders.

In case of analyses please report witch software had been used. Additionally, effect sizes can be computed based on fixed and or random model. Which one was used?

I cannot find table 1 and 2

Publication bias should be reported, I recommend funnel plot and Duval and Tweedie’s trim and fill method.

Egger, M., Smith, G. D., Schneider, M., & Minder, C. (1997). Bias in meta-analysis detected by a simple, graphical test. BMJ, 315, 629–634. https://doi.org/10.1136/bmj.315.7109.629

Duval, S., & Tweedie, R. (2000). Trim and fill: A simple funnel-plot–based method of testing and adjusting for publication bias in meta-analysis. Biometrics, 56, 455–463. https://doi.org/10.1111/j.0006-341X.2000.00455.x

Author Response

This study investigates the effects of mindfulness-based interventions on ADHD symptoms. Although this is an important topic, the present study suffering lacks novelty, as several studies on similar topics have been published.

  1. Cairncross, M., & Miller, C. J. (2020). The effectiveness of mindfulness-based therapies for ADHD: A meta-analytic review. Journal of attention disorders, 24(5), 627-643.
  2. Chimiklis, A. L., Dahl, V., Spears, A. P., Goss, K., Fogarty, K., & Chacko, A. (2018). Yoga, mindfulness, and meditation interventions for youth with ADHD: Systematic review and meta-analysis. Journal of Child and Family Studies, 27(10), 3155-3168.
  3. Vekety, B., Logemann, H. A., & Takacs, Z. K. (2021). The effect of mindfulness-based interventions on inattentive and hyperactive–impulsive behavior in childhood: A meta-analysis. International Journal of Behavioral Development, 45(2), 133-145.
  4. Xue, J., Zhang, Y., & Huang, Y. (2019). A meta-analytic investigation of the impact of mindfulness-based interventions on ADHD symptoms. Medicine98(23).

So I do not recommend to publish it in the International Journal of Environmental Research and Public Health.

Response: Although there were several publications of meta-analysis of mindfulness-based intervention, this was the first meta-analysis that only included randomized controlled trials. The findings provided evidence regarding the effects of mindfulness-based intervention for children and adolescents with ADHD.

My comments are the following:

The authors mention to include not just mindfulness but yoga and Taiichi interventions. It would be better to treat them in separate analyses.

Response: There were no Taiichi included in this meta-analysis. There were two studies using yoga (Cohen et al., 2018; Jensen & Kenny, 2004) included in this meta-analysis. However, Cohen and colleagues (2018) did not provide data for meta-analytic synthesis. We have done sensitivity analysis by excluding each study (Jensen & Kenny, 2004) and found stable results.

A big benefit is to provide methodological quality assessment was done. I recommend to compare the effect fas found in good vs fair quality studies.

Response: We used scores of methodological quality assessment as moderator to run meta-regression with the effects of mindfulness-based intervention. The results showed that the methodological rating had a small and negative association with effect sizes of ADHD symptoms and internalizing behavioral problem respectively (Page 10, Line 2-4; 8-10). Studies with better quality had less effect sizes and the magnitude of association was small.

Data extraction would have been better if it was performed by two independent coders.

Response: The first author extracted and tabulated the relevant data from the studies, and the second author double-checked the information.

In case of analyses please report witch software had been used. Additionally, effect sizes can be computed based on fixed and or random model. Which one was used?

Response: The section ”Statistical Analysis” was added in the methods. The software used and model for compute effect sizes are presented in the section “Statistical Analysis.” (Page 4, Line 19 – Page 5, Line 8)

I cannot find table 1 and 2

Response: Tables 1, 2, and 3 were added.

Publication bias should be reported, I recommend funnel plot and Duval and Tweedie’s trim and fill method.

Response: Publication bias was reported. Funnel plot, the Begg and Mazumdar test and the Duval and Tweedie’s trim and fill method were used. (Page 5, Line 5-7; Page 8, Line 6-10; 19-23; 34-38; Page 9, Line 11-15).

Egger, M., Smith, G. D., Schneider, M., & Minder, C. (1997). Bias in meta-analysis detected by a simple, graphical test. BMJ, 315, 629–634. https://doi.org/10.1136/bmj.315.7109.629

Duval, S., & Tweedie, R. (2000). Trim and fill: A simple funnel-plot–based method of testing and adjusting for publication bias in meta-analysis. Biometrics, 56, 455–463. https://doi.org/10.1111/j.0006-341X.2000.00455.x

Round 2

Reviewer 2 Report

However the authors have modified their manuscript according to the comments left by the reviewers, I do not find this paper qualified for publication.

Author Response

We thank the reviewer for the graceful encouragement. "However the authors have modified their manuscript according to the comments left by the reviewers.